# Crossing complexity of space-filling curves reveals entanglement of S-phase DNA

**Nick Kinney**[1,2]*, **Molly Hickman**[3], **Ramu Anandakrishnan**[1,2], **Harold R. Garner**[1,2]

**1** Edward Via College of Osteopathic Medicine, Blacksburg, VA, United States of America, **2** Gibbs Cancer Center & Research Institute, Spartanburg, SC, United States of America, **3** Department of Computer Science, Virginia Tech, Blacksburg, VA, United States of America

* nkinney06@gmail.com

**Data Availability Statement:** All relevant data are within the manuscript and its Supporting Information files.

**Funding:** The authors received no specific funding for this work.

## Abstract

Space-filling curves have been used for decades to study the folding principles of globular proteins, compact polymers, and chromatin. Formally, space-filling curves trace a single circuit through a set of points (x,y,z); informally, they correspond to a polymer melt. Although not quite a melt, the folding principles of Human chromatin are likened to the Hilbert curve: a type of space-filling curve. Hilbert-like curves in general make biologically compelling models of chromatin; in particular, they lack knots which facilitates chromatin folding, unfolding, and easy access to genes. Knot complexity has been intensely studied with the aid of Alexander polynomials; however, the approach does not generalize well to cases of more than one chromosome. Crossing complexity is an understudied alternative better suited for quantifying entanglement between chromosomes. Do Hilbert-like configurations limit crossing complexity between chromosomes? How does crossing complexity for Hilbert-like configurations compare to equilibrium configurations? To address these questions, we extend the Mansfield algorithm to enable sampling of Hilbert-like space filling curves on a simple cubic lattice. We use the extended algorithm to generate equilibrium, intermediate, and Hilbert-like configurational ensembles and compute crossing complexity between curves (chromosomes) in each configurational snapshot. Our main results are twofold: (a) Hilbert-like configurations limit entanglement between chromosomes and (b) Hilbert-like configurations do not limit entanglement in a model of S-phase DNA. Our second result is particularly surprising yet easily rationalized with a geometric argument. We explore ergodicity of the extended algorithm and discuss our results in the context of more sophisticated models of chromatin.

## Introduction

In eukaryotes, chromatin adopts highly folded configurations within the boundaries of the nucleus; however, the principles that govern chromatin folding are not fully understood. Progress has been made using a combination of experimental observations explained by computational models of various complexity. Models based on space filling curves (SFC) are among the simplest. Given a set of points (x,y,z), a SFC visits each once and only once; consequently, they are Hamiltonian and sometimes referred to as Hamiltonian paths. Lattice SFCs generally refer

**Competing interests:** No, the authors have declared that no competing interests exist.

to a special case with two additional stipulations: (a) the points are arranged in a three-dimensional grid, (b) each step in the path visits an adjacent point on the lattice. SFCs are not unique; on the contrary, the partition function scaling grows exponentially with curve length (L): $Z \simeq 2.135^L$ for equilibrium curves [1] and $Z \simeq 1.357^L$ for Hilbert-like curves [2]. Fortunately, algorithms exist for sampling lattice SFCs; unbiased sampling has been demonstrated with the Mansfield algorithm [3].

Techniques designed to sample lattice SFCs can be traced back to the 1980's. Key algorithms were proposed by Ramakrishnan, Pekny, and Caruthers [4]; Lua, Borovinskiy, and Grosberg [5]; and Marc Mansfield [3]. The Mansfield algorithm was the first to show ergodicity exhaustively on small lattices and non-exhaustively on larger lattices [3]. Mansfield also introduced an algorithm to handle multiple curves at high volume fraction [6].

Globular proteins, compact polymers, and chromatin are undoubtedly more complex than lattice SFCs. In fact, any real system of polymers has a myriad of additional details. It is important to note that no amount of detail can abolish all statistical properties of the system. For example, the 3D distance between any two monomers ($R$) cannot exceed their 1D separation along the polymer backbone ($s$). Mathematically this constraint can be expressed as an inequality ($R <= s$) satisfied regardless of the system details. Thus, simple lattice SFCs share at least some properties of much more complex systems and can provide answers to general questions about those systems. Typically, these questions are statistical in nature. The aforementioned inequality ($R <= s$) may be framed as such a question: what is the average 3D distance between monomers? Answers to this question often fit ensembles of polymers to a power law ($R = s^v$) where the scaling exponent ($v$) reflects the degree of polymer folding [7, 8].

Space filing curves, compact polymers, and chromatin are related by a mutual correspondence to the state of polymers in a melt. Broadly speaking, this state corresponds to that of dense intermingled blobs, but, one in which polymers are still in thermal motion [9]. Although chromatin is not quite a melt [8], such a dense state with the potential to form tangles and knots is a conceivable hazard to chromatin folding and unfolding. However, simulations suggest that knots in melts are fewer than expected [10]; in any case, human chromosomes are probably not in equilibrium [11]. Chromatin is believed to fold into quasi-equilibrium configurations that limit entanglement between and within chromosomes [8, 11, 12]. A model consistent with this observation was originally proposed by Grosberg in 1988 based on the concept of the crumpled globule [13]. The crumpled globule is one of the key theoretical shapes in the field of genome organization [14].

A crumpled globule can be made by recursively folding an elongated polymer [13]. First, a fiber-of-crumples is formed by introducing small folds along the polymer backbone. Larger folds are introduced along the fiber-of-crumples and the process reiterates. The result is a structure that possesses self-similarity on all length scales [13]; i.e. contiguous segments of a crumpled globule are also crumpled. In fact, self-similarity of the crumpled globule has led to a more common name: the fractal globule [12, 13]. The crumpled (fractal) globule has a distinct scaling exponent ($v$) and several properties that make it an attractive model of chromatin folding. In particular, its scaling exponent does not vanish for large genomic distances, which reflects the configuration's territorial organization [12]. Moreover, the crumpled globule clearly limits chromosome entanglement which makes it easy to fold and unfold [14].

Metrics for studying polymer entanglement are limited. Most studies advocate the use of Alexander polynomials [15–19]; recent studies have introduced crossing complexity as an alternative [20, 21]. Briefly, Alexander polynomials are knot invariants useful for identifying equivalent knots and quantifying knot complexity. Use of Alexander polynomials has led to biologically significant results; for instance, the presence of knots may stabilize a protein's native folded state [22, 23]. A few studies of genome organization have adopted the idea for

detecting entangled chromosomes [8, 12, 14, 24, 25]. Indeed, knot complexity suggests that the crumpled globule limits chromosome entanglement [14]. On the other hand, it has been suggested that knot complexity is insufficient to distinguish unknotted equilibrium configurations from Hilbert-like configurations [15]. Regardless, knot complexity continues to be the gold standard for quantifying chromosome self-entanglement. Attempts have been made to quantify entanglement between polymers (chromosomes) with variants of the Gauss linking number [26]. The linking number increases with polymer length and density [27, 28]. Crossing complexity is a simple yet understudied alternative well-suited for quantifying entanglement between chromosomes [20, 21]. Crossing complexity uses putative translations to quantify the process of physically separating two chromosomes.

This work is divided into three sections. The first section introduces a generalized Mansfield algorithm for sampling lattice SFCs. The generalized algorithm enables full control over the number of SFCs on the lattice and partial control over their power law ($R = s^v$). This in turn enables sampling of Hilbert-like configurations. The second section begins with a review of crossing complexity for quantifying polymer entanglement and also introduces novel visualizations for this entanglement. Crossing complexity is then compared for three ensembles of lattice SFCs: equilibrium, intermediate, and Hilbert-like. We show that doubled curves (akin to S-phase DNA) have the same crossing complexity regardless of how they are folded. The third section revisits the statistics of configurations sampled with the new lattice SFC algorithm; our results suggest they are ergodic. In support of recent theoretical results, Hilbert-like configurations simplify crossing complexity for two different chromosomes. However, Hilbert-like configurations do not simplify crossing complexity in a lattice SFC model of S-phase DNA. We conclude that some states of DNA are more tangled than we thought.

## Results

### Multiple lattice space filling curves efficiently sampled using a generalized Mansfield algorithm

Techniques designed to sample lattice SFCs–specifically those developed by Marc Mansfield–include variations for single and multiple chains. Both variations of the Mansfield algorithm rearrange chains using a "backbite" maneuver (see methods). The Mansfield algorithm for multiple chains also uses an "end attack" maneuver (see methods). Only three algorithmic insertions convert iterations of the single chain algorithm to the multi-chain algorithm. These insertions are made clear in Fig 1 without further discussion.

We introduce a "nest and replace" chain rearrangement to partially control chain end-to-end scaling. The simplest case in 2-dimensions is briefly described here with additional details in methods (see Fig 2). Begin by exhaustively precomputing 2x2 SFCs: it is important not to eliminate paths identical by rotation. Next, use the precomputed 2x2 paths to replace vertices of an existing lattice SFC: green curve in Fig 2. Spatial separation of chain endpoints ($R$) will approximately double; the number of bonds ($s$) will increase by a factor of 4; and the scaling exponent ($v$) will necessarily decrease. To see this, consider the power law: $R = s^v$. Successive iterations will further decrease the scaling exponent. The approach easily generalizes to higher dimensions. In the results that follow, we use this approach–in 3 dimensions–to generate and study ensembles of equilibrium, intermediate, and Hilbert-like configurations (see methods for details).

### Crossing complexity for multiple space filling curves

Knot complexity has been intensely studied with the aid of Alexander polynomials; and, continues to be the gold standard for quantifying self-entanglements. However, the approach does

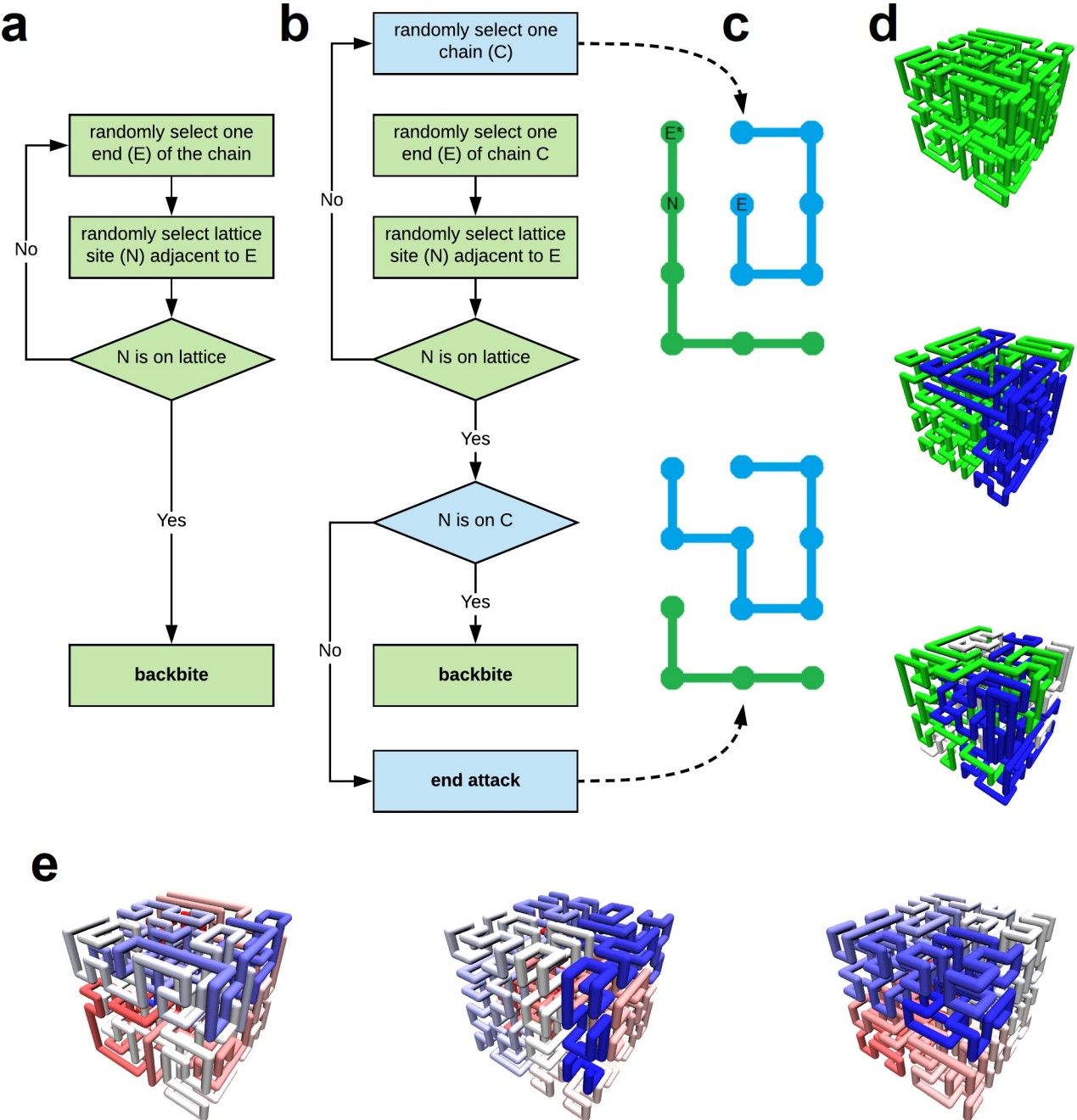

**Fig 1. Algorithm for generating multiple lattice Space Filling Curves (SFC) on a simple cubic lattice. a,** Mansfield algorithm for generating single lattice SFCs. **b,** Mansfield algorithm for generating multiple lattice SFCs with algorithmic insertions in blue. **c,** Schematic of the end-attack chain rearrangement maneuver. **d,** Configurational snapshots of one, two, and three SFCs on a simple cubic lattice, respectively. **e,** Configurational snapshots of equilibrium, intermediate, and Hilbert-like SFCs on a cubic lattice, respectively. Each lattice has one chain; blue to red colormap emphasizes differences in scaling. Configurations are generated with novel steps that partially control chain end-to-end scaling (see methods).

not generalize well to cases of more than one chromosome. Crossing complexity is an under-studied alternative better suited for quantifying entanglement between chromosomes. Here we investigate crossing complexity in three ensembles of lattice SFCs: equilibrium, intermediate, and Hilbert-like. Details of crossing complexity can be found elsewhere [20, 21]; nevertheless, we briefly reiterate the concept for two curves (Fig 3). Putative translations are applied over a

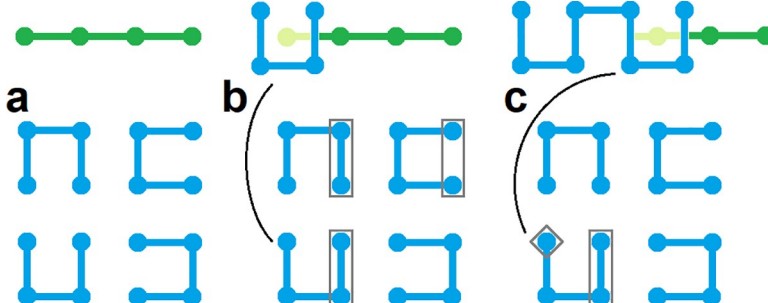

**Fig 2. Computational details of nest and replace chain rearrangements.** Monomers in the green lattice SFC (top left) are replaced sequentially–from end to end–by configurational snapshots of 2x2 lattice SFCs (blue). **a,** Begin by exhaustively computing all 2x2 lattice SFCs. **b,** End-wall orientation of valid replacement paths (grey boxes) must match the bond orientation of the monomer (to be replaced) and its neighbor (next to be replaced). **c,** Start point of valid replacement paths must be adjacent to the endpoint of the previous replacement (grey diamond). Valid replacements are selected at random. Iterative steps (b) and (c) are repeated for all monomers. The procedure is trivial to generalize for three dimensions.

**a**

**b**

**c**

**d**

**e**

## number of crossings

**Fig 3. Crossing complexity for two curves on a simple cubic lattice. a,** Two SFCs on a simple 16x16x16 cubic lattice. **b,** Putative translations are applied over a set of test directions that evenly cover the space $S^2$ (spherical surface); chain crossings are enumerated and color coded for each direction. **c,** Enumerated chain crossings mapped onto the unit sphere. **d,** Enumerated chain crossings mapped onto the unit cube. **e,** Mercator projection of chain crossings.

set of test directions that evenly cover the space $S^2$ (spherical surface). The two curves are separated–to completion–along the axis of each test direction while counting their mutual crossings (Fig 3B). Statistics of the enumerated chain crossings are used to infer their degree of entanglement.

We construct three configurational ensembles (equilibrium, intermediate, and Hilbert-like) using a generalized Mansfield algorithm with nest and replace chain rearrangements (details in methods). Each ensemble consists of 800 uncorrelated configurational snapshots. Each snapshot is generated on a 16x16x16 lattice with two chains; i.e. each chain occupies 2048 points. Chain crossings are enumerated for 1000 test directions and pooled for all configurations in the three ensembles, respectively. The distribution of enumerated chain crossings clearly differs for each ensemble (Fig 4D and 4E). The Hilbert-like ensemble produces an asymmetric (non-normal) distribution with average number of crossings lower than equilibrium and intermediate ensembles, respectively. Roughly speaking, Hilbert-like curves require fewer chain crossings to pull apart; this interpretation is consistent with easy unfolding of Hilbert-like configurations in simulation [12] and their lack of entanglement inferred from knot complexity [14]. See Fig S27 of the latter reference.

In addition to ensemble differences, links to chromatin folding are bolstered by considering individual snapshots (loosely representative of individual cell nuclei). In particular, is crossing complexity sufficient to discriminates equilibrium, intermediate, and Hilbert-like configurational snapshots? Or simply, what is the snapshot (cell) variability? To answer this question, we reinvestigate the configurational snapshots in each ensemble. For each snapshot we use a Kolmogorov-Smirnov test to classify the distribution of enumerated chain crossings as equilibrium, intermediate, or Hilbert-like (details in methods). Less than 1% (.625%) of equilibrium snapshots are classified as Hilbert-like (Fig 4F); similarly, less than 2% (1.63%) of Hilbert-like snapshots are classified as equilibrium (Fig 4H). We conclude that crossing complexity does discriminate equilibrium and Hilbert-like configurational snapshots.

## High crossing complexity in a model of S-phase DNA

Crossing complexity is also relevant for states of DNA that arise in S-phase after replication. First principles suggest this state consists of two strands (template and newly synthesized) juxtaposed in register. In subsequent phases the two strands separate. We hypothesize that the configuration of replicated S-phase DNA influences the number of crossings needed for strand separation. We test this hypothesis in two steps. First, we construct snapshots of S-phase DNA (described below). Then we compare crossing complexity for equilibrium, intermediate, and Hilbert-like ensembles. The results–which contradict our hypothesis–lead to one of the main conclusions of this study: the crossing complexity of doubled space-filling curves is the same regardless of how they are folded.

We construct ensembles of S-phase DNA for three classes of lattice SFCs: equilibrium, intermediate, and Hilbert-like. Each ensemble consists of 800 uncorrelated configurational snapshots. Each snapshot is generated in two steps. First, a single SFC is generated on a 16x16x16 lattice. Next, the curve is replicated in place akin to S-phase DNA; for clarity, a schematic is shown for a smaller 4x4x4 curve (Fig 5A–5C). Chain crossings are enumerated for 1000 test directions and pooled for all configurations in the three ensembles, respectively. Distributions of enumerated chain crossings for each ensemble are almost identical for intermediate and Hilbert-like ensembles. In other words, Hilbert-like configurations do not reduce the chain crossings required to pull apart two doubled curves (chromosomes) in S-phase.

Biological interpretation of our results must be made with caution. The variance within each distribution clearly exceeds any difference in means. This seems to negate the folding

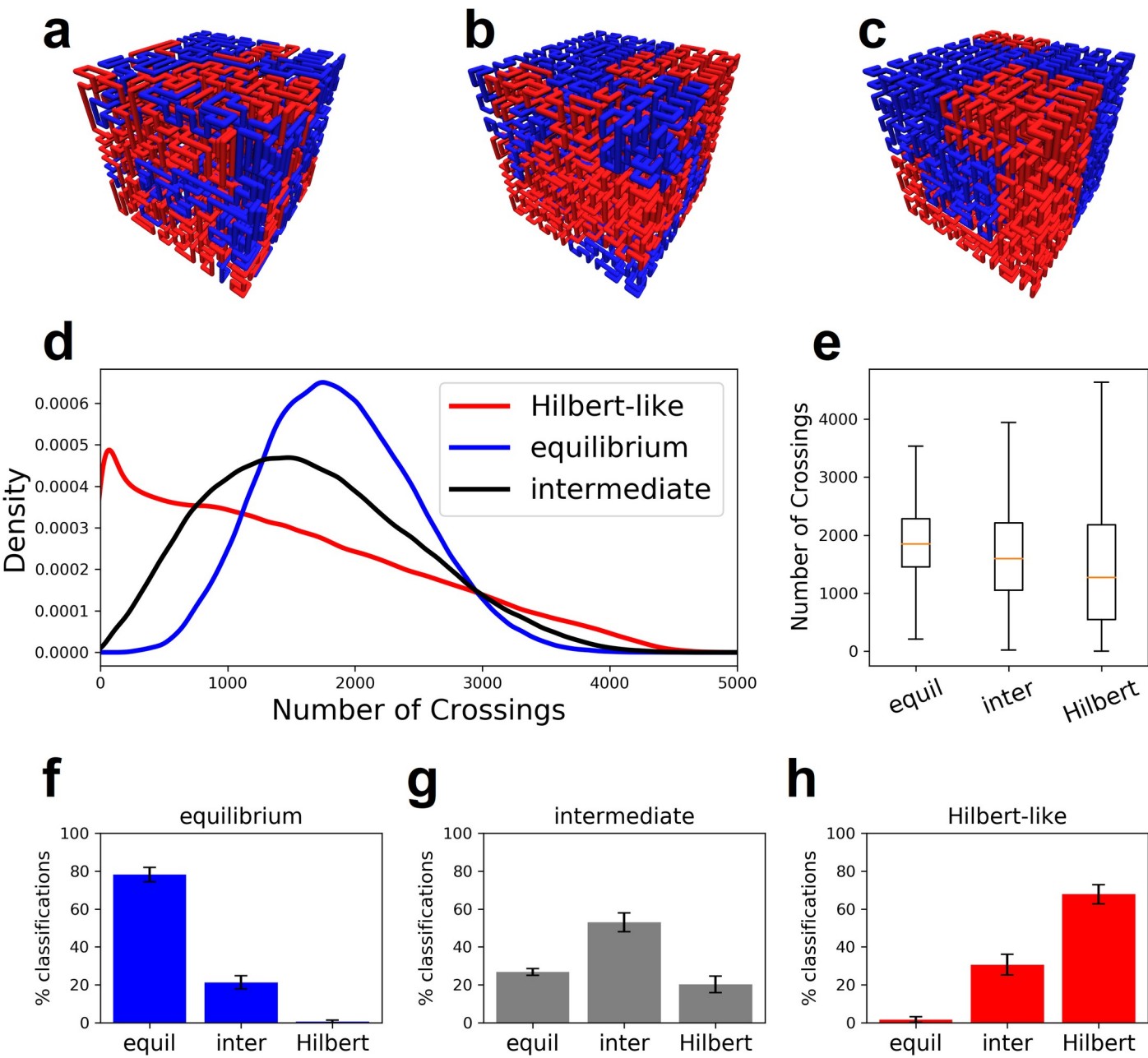

**Fig 4. Crossing complexity for two Lattice space filling curves on a simple cubic lattice. a-c,** Configurational snapshots of equilibrium, intermediate, and Hilbert-like curves, respectively. **d-e,** The distribution of crossing complexity enumerated for 1000 test directions in ensembles of equilibrium, intermediate, and Hilbert-like curves. Average crossing complexity is lowest for Hilbert-like curves. **f-h,** Crossing complexity partially distinguishes between snapshots of each ensemble. We used a Kolmogorov-Smirnov test to classify snapshots in each ensemble. In each case most of the snapshots classify into the correct ensemble (details in methods).

principles of each ensemble altogether; i.e. test directions matter more. We conclude that Hilbert-like configurations may not mitigate the number of chain crossings required to physically separate replicated S-phase DNA. Biologically favorable properties of these configurations–lack of knots and tangles–may simply be more important for chromosome folding and unfolding in interphase. This interpretation necessitates a need for more sophisticated models of S-phase DNA.

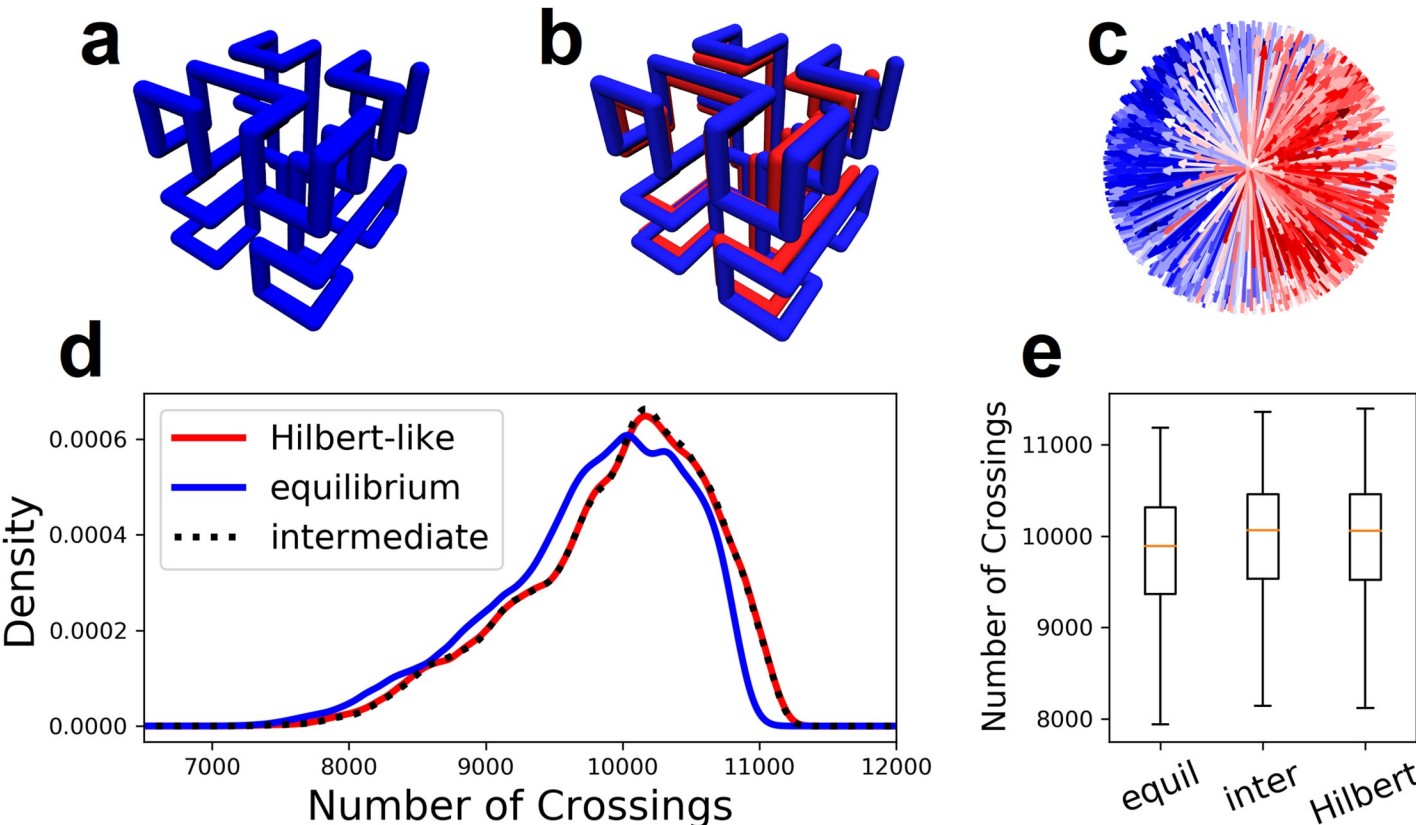

**Fig 5. High crossing complexity in a lattice Space Filling Curve (SFC) model of S-phase DNA. a,** Begin with a single lattice SFC; we consider equilibrium, intermediate, and Hilbert-like configurations. **b,** Replicate the Lattice SFC in place akin to S-phase DNA; the red and blue curves (chromosomes) remain juxtaposed in in register. **c,** Compute the crossing complexity by enumerating chain crossings over a set of test directions that evenly cover the space $S^2$. **d,** Distribution of enumerated chain crossings for each ensemble. **e,** Average number of chain crossings is lowest in the equilibrium ensemble and comparable in the intermediate and Hilbert-like ensembles.

## Crossing complexity of doubled curves rationalized through geometry

Surprisingly, the doubled (S-phase) configurations appear to produce the same distribution of chain crossings regardless of their underlying folding principles (equilibrium, intermediate, or Hilbert-like). We rationalize this observation with a simple geometric argument.

First, consider projecting layers of a space-filling curve in the plane perpendicular to one axis; for example, the space filling curve in Fig 6A and its projections in Fig 6B–6E. In this case, bonds parallel to each projection number 11, 9, 10, and 11, respectively (Fig 6B–6E). In fact, the parallel bonds in each projection are predictable. Simply two thirds of the 63 total bonds are divided into four layers. Therefore–assuming no spatial anisotropy–we expect ten or eleven bonds in each projection regardless of folding principles.

Next, consider the doubled curve (red Fig 6F). Follow a single bond through each layer (red in Fig 6G–6J). The probability of chain crossing is proportional to the number of bonds in each layer. Regardless of how the blue curve is folded, each layer has 24 positions for ten or eleven bonds. Consequently, we expect the red segment to produce ~2 chain crossings as it traverses layers of the blue curve (Fig 6G and 6J). Crossing complexity simply sums over each segment, therefore, is not expected to depend on folding principles for the doubled curves.

What if we follow the red segment in skew directions, i.e. directions not perpendicular to the plane of the blue curve? Even in these cases (which are the majority) our argument remains

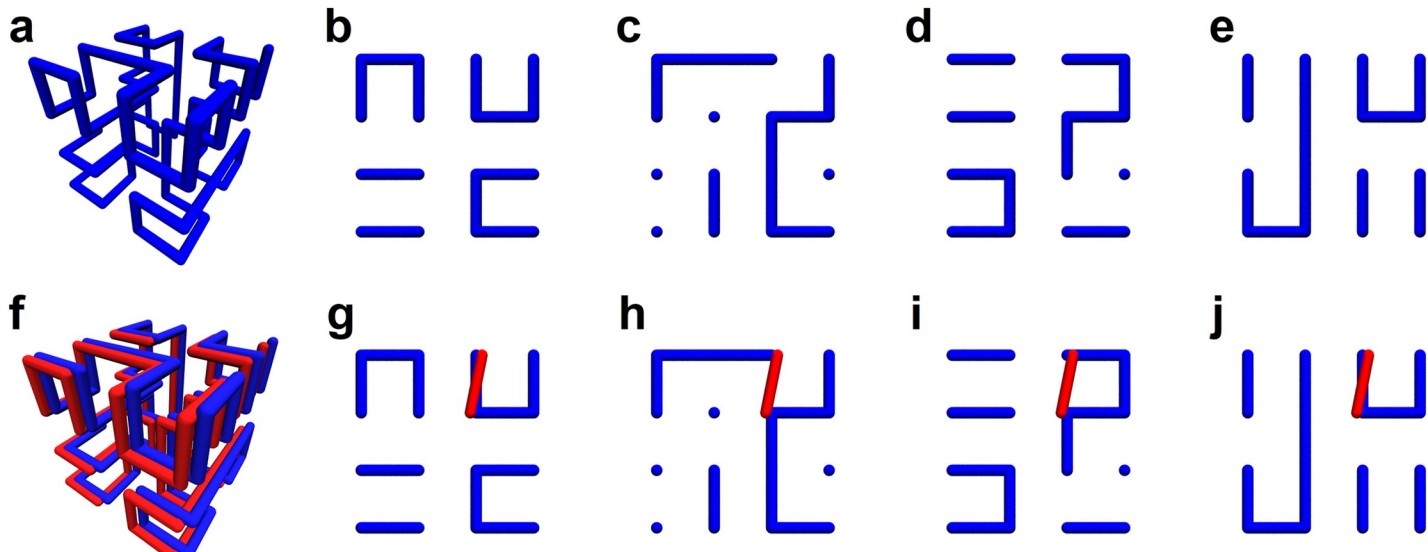

**Fig 6. A simple geometric argument suggests that doubled space-filling curves produce the same distribution of enumerated chain crossings regardless of folding principles. a,** Space filling curve on a 4x4x4 lattice. **b-e,** Four layers of the space filling curve projected along one axis. **f,** A doubled 4x4x4 space filling curve. **g-j,** A single segment of the doubled curve followed through each layer of the blue curve. Chain crossings occur in panels g and j.

fundamentally unchanged. Bonds in the blue curve are sufficiently random that the probability of chain crossing for each red segment is the same regardless of folding principles.

## Ergodic sampling of lattice space filling curves

We revisit the algorithm used throughout this work to address questions of ergodicity. To reiterate, the algorithm uses two Monte Carlo moves–backbites and end-attacks–to efficiently sample multiple SFCs on a simple cubic lattice. In fact, Mansfield was first to hypothesized that these two moves are sufficient for unbiased sampling; a third rearrangement maneuver (bond flip) was only included in his original work to improve statistics [6]. We support this hypothesis by checking statistics for an ensemble of 1000 uncorrelated configurational snapshots. Each snapshot is generated on an 8x8x8 lattice with two chains; i.e. each chain occupies 256 points.

First, we investigate probabilities of chain endpoint occupancy, i.e. how often chain ends occupy each lattice site. We find the probability of chain ends at each lattice site distributed around the inverse of the lattice size (Fig 7A). This seems reasonable for an unbiased configuration (it probably is in our case); however, it is not the case for every lattice. For example, the 3x3x3 lattice has well known parity rules that exclude chain endpoints from occupying even-numbered (center of each edge) lattice sites. More complicated parity rules have been shown for second order Hilbert curves [2]. Thus, it should not be assumed a priori that unbiased configurations produce a normal distribution of endpoints. Simply put, the normal distribution of chain endpoints in our configurations does not obviously rule out ergodicity of the multi-chain algorithm used throughout this work. To have more confidence in this result we check the distribution of endpoints for single chain configurations on a 6x6x6 lattice (each chain occupies 216 points). Note that the Mansfield algorithm for single chains is very likely ergodic [3]. We enumerate 1000 single chain configurations on a 6x6x6 lattice and find a similar normal distribution of chain end occupancy (S1 Fig).

We check chain torsion using the triple product for all sets of three consecutive bonds. Briefly, the triple product produces a scalar in the range -1 to 1; negative values indicate left-

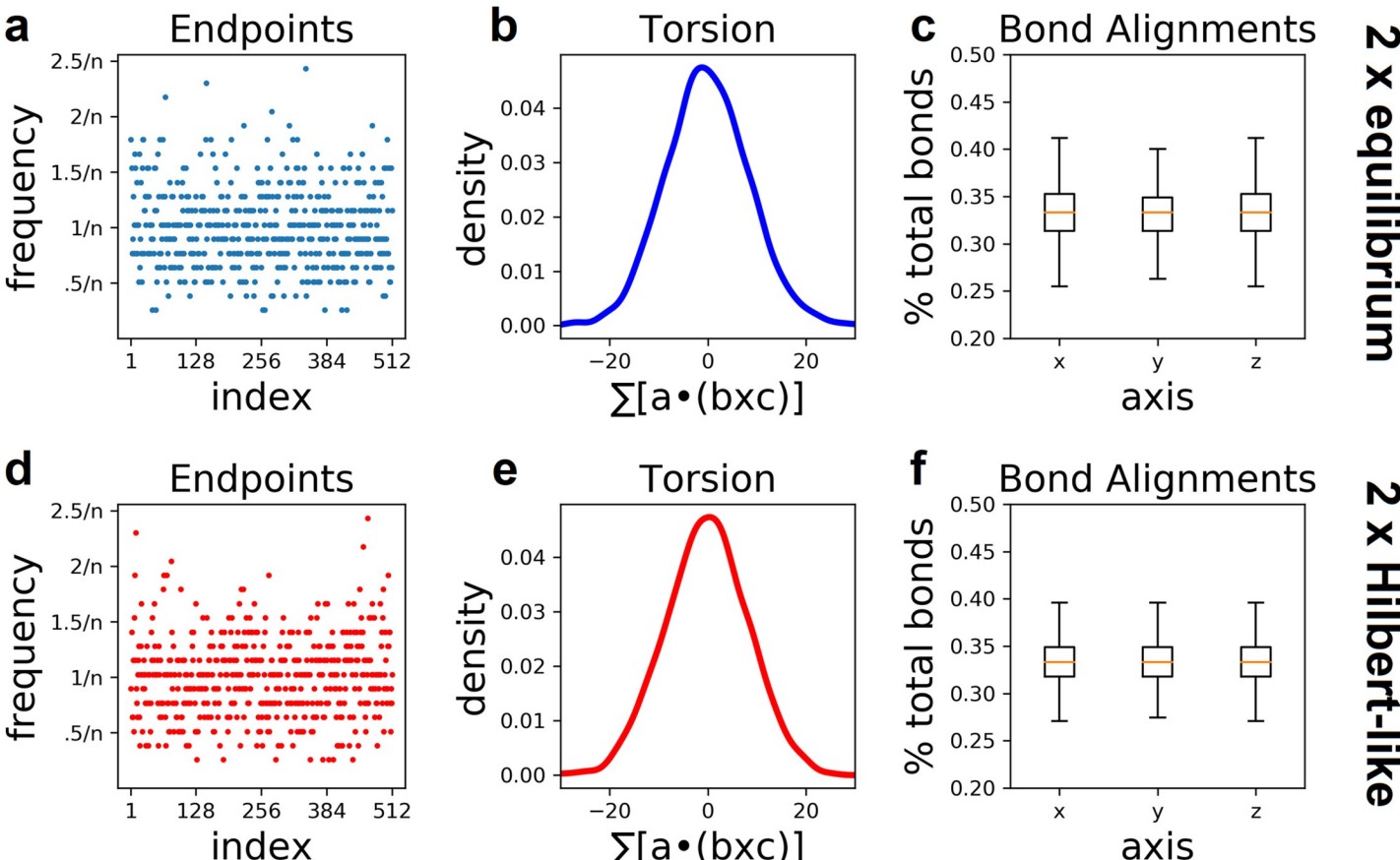

**Fig 7. Statistics of lattice space filling curves suggest they are ergodic. Top row,** Results for equilibrium configurations. **Bottom row,** Results for Hilbert-like configurations. **a and d,** Probability for chain ends at each lattice site are distributed around the expected value: inverse of the lattice size (n). **b and e,** Distribution of total chain torsion (measured with the triple product) has zero mean. **c and f,** Bonds for each chain have equal probability of alignment along the x, y, and z axes. These findings suggest–but do not prove–that configurations are sampled without bias.

handed torsion while positive values indicate right-handed torsion (details in methods). We compute the total torsion (sum of triple products) for all chains in the aforementioned configurational ensemble. The distribution of total torsion is symmetric around zero (Fig 7B). This result seems reasonable for an unbiased sample. To see this, consider a hypothetical lattice configuration with a degree of right-handed torsion. There must be a "mirror image" configuration with the same degree of left-handed torsion. In fact, for every configuration with right-handed torsion there should be a configuration with equal left-handed torsion. Thus, we expect the distribution of torsion in an unbiased ensemble to be symmetric (not necessarily bell-shaped) around zero.

We check for random orientation of bonds in each chain; that is, bonds should be random with respect to their alignment along the x, y, or z axis. We find an equal probability for each orientation (Fig 7C). We double-check (as above) by making the same observation in an unbiased sample of 1000 configurations on a 6x6x6 lattice (S1 Fig).

These findings suggest–but do not prove–that configurations are sampled without bias. Numerous statistics not considered here could be used to further support or refute our findings. In many cases the expectations for an unbiased sample are highly non-trivial or computationally unfeasible. Future work will build on the results presented here.

### Hilbert-like configurations are evenly sampled

Questions of ergodicity are even harder to address for algorithms that sample Hilbert-like configurations. Indeed, these are non-equilibrium configurations: a small subset of all possible configurations. To what extent is the subset of Hilbert-like configurations sampled without bias? We address this question by checking statistics for an ensemble of 1000 uncorrelated configurational snapshots. Each snapshot is generated on an 8x8x8 lattice with two Hilbert-like chains; each chain occupies 256 points.

Three statistical properties of the Hilbert-like ensemble suggest configurations are sampled without bias. First, frequency of chain end occupancy is distributed around the expected value: inverse of the lattice size (n) (Fig 7D). We reiterate that this does not obviously rule out ergodicity of the algorithm used throughout this work. Second, the distribution of total torsion is symmetric around zero (Fig 7E). Third, bond orientations are random with respect to alignment along the x, y, and z axis (Fig 7F). These findings suggest that our algorithm evenly samples the subset of Hilbert-like configurations with respect to these chain statistics; however, it is conceivable that bias exists for chain statistics not examined here.

## Discussion

Lattice SFCs have been used for decades to study the fundamental properties of polymers. The folding of Human DNA has been likened to the Hilbert curve: a special case of the lattice SFC [12, 14]. Building on this result, a recent study proposed that "tessellation" globules are more accurate over larger genomic distances [29]. These advances alone reiterate that lattice SFCs continue to be useful first line models. A main contribution of this work is a generalized Mansfield algorithm for sampling lattice SFCs [3, 6]. The algorithm enables full control over the number of SFCs on the lattice and partial control over their power law ($R = s^v$). Further contributions of this work notwithstanding, the algorithm has a particularly wide range of applications. Foremost, it can be used to generate computational ensembles of paths that in turn represent chromosomes, proteins, or other dense polymer systems; and, configurational snapshots can be used as starting points for simulations. Simple modifications of the approach can be used to control chain length, system density, and other relevant features of the system.

Additional contributions of this work stem from an investigation of crossing complexity for SFCs. The main message is twofold: (a) Hilbert-like configurations limit crossing complexity between chromosomes and (b) Hilbert-like configurations do not limit crossing complexity once doubled akin to S-phase DNA. Despite these contributions, crossing complexity remains understudied compared to related metrics such as knot complexity ($\Delta$), surface smoothness ($\beta$), and fractal dimension. Here surface smoothness refers to the monomers of one curve in contact with other curves governed by the exponent $\beta$: $n_{surf} \sim N^\beta$. Hilbert-like curves are characterized by $\beta = 2/3$, knot complexity $\Delta(-1) = 1$, fractal dimension = 3, and modal crossing complexity (see results) approximately zero. It is natural to seek a quantitative comparison of these metrics. We hypothesize the correlation is strongest for crossing complexity and surface smoothness; however, a thorough study is beyond the scope of this work. In anticipation of more widespread use, we parallelize the crossing complexity using open MPI and develop novel visualizations over the space $S^2$ (spherical surface). Our code is freely available online.

The de facto model of human chromatin–the crumpled globule–has been shown to mitigate knot complexity and crossing complexity in silico, and empirical evidence indirectly supports these theoretical findings [12, 14, 30]. For example, real chromosomes clearly possess territories that only intermingle on their boundaries. Our results suggest that crumpled (fractal) globules possess asymmetric crossing complexity. This builds on what is known about the crumpled globule and reinforces consensus that they lack entanglement. A more interesting

result stems from a simple model of S-phase DNA. Our results suggest that replicated DNA in S-phase is entangled regardless of its folding principles. Once replicated, crumpled globules may even possess higher average crossing complexity than equilibrium globules. Work needs to be done to repeat this result in more sophisticated models of chromatin folding; however, it seems unlikely that model details will have a significant effect on the landscape of crossing complexity.

Our results present an apparent contradiction: Hilbert-like configurations limit entanglement between each other but not once replicated in place akin to S-phase DNA. We offer an interpretation that bolsters the significance of chromosome territories. Perhaps knots and chain crossings do not hinder the otherwise normal functioning of chromatin; indeed, cells have topoisomerase for this purpose. We propose that territorial organization of Hilbert-like configurations may be their more significant biological feature. Indeed, territories are a unique property of the crumpled globule–more so than lack of knots–and have been observed in a wide range of organisms: yeast, human, fruit fly, mouse, and Arabidopsis [31].

This work sets the stage for numerous future lines of inquiry. First, we suggest a more thorough investigation of ergodicity for the generalized Mansfield algorithm. In Mansfield's original paper questions of ergodicity were tested using exact enumeration on small lattices and non-exhaustive enumeration on larger lattices [3]. Our approach permits multiple paths; consequently, exhaustive enumeration may be unfeasible on any size lattice. We show that paths are flexible and torsion free; however, a better test of ergodicity needs a sampling procedure that so far has not been developed.

The simplicity of SFCs is both a strength and a weakness. Simple construction leads to reproducible results at the cost of biological realism.

A clear omission in this work is the presence of topological associated domains (TADs). TADs support a hypothesis that the mammalian genome is folded into tight-knit globular chromatin connected by linear chromatin boundaries [32]. Are TADs a detail that co-exists within the more salient folding principles of the crumpled globule? If so, our results should be reproducible in models that include TADs regardless of their omission here. Is the TAD model of chromatin fundamentally different than the crumpled globule model? If that is the case, crossing complexity and knot complexity should be examined in a TAD based model. Reality is probably somewhere in between. In the kilobase to megabase range, Hi-C contact maps are consistent with the crumpled globule. Deviations from the crumpled globule arise beyond several megabases and within TADs [29]. Understanding crossing complexity in more sophisticated models will be pursued in a future work.

## Materials and methods

### Chain rearrangement maneuvers

We use two algorithms to sample equilibrium and Hilbert-like SFCs, respectively. Equilibrium paths are generated using the variation of an algorithm proposed by Mansfield. Briefly, two chain rearrangement maneuvers are used to generate configurational snapshots: end attacks and backbites. Mansfield's original algorithm used a third chain rearrangement maneuver: bond flips. We introduce a fourth chain rearrangement to control end-to-end scaling and generate Hilbert-like configurations: nest and replace. Details for each of the four chain rearrangement maneuvers are provided here; computational details are provided in the next sections.

a. End attack is an intermolecular chain rearrangement in which a bond is broken and annealed to the end of a different chain (Fig 8A). Typically, two valid rearrangements are possible depending on which bond is broken; in our implementation this choice is made at random.

b. Backbite is an intramolecular chain rearrangement in which a bond is broken and annealed to the end of the same chain (Fig 8B). Just like the end attack, two valid rearrangements are possible depending on which bond is broken; however, one always leads to cyclization. In our implementation all cyclization is prohibited.

c. Bond flip is an intermolecular chain rearrangement in which two parallel bonds are broken and reformed in perpendicular orientation (Fig 8C). Chain ends stay fixed. Bond flips are not included in our implementation.

d. Nest and replace is a rearrangement in which a single monomer is replaced by an entire configurational snapshot (Fig 8D). In our implementation, all replacements are precomputed and correspond to the set of all possible configurations of 2x2x2 lattice SFCs. Valid replacements are made at random: computational details are provided below.

## Computational pitfalls when generating snapshots

Combinations of chain rearrangement maneuvers can be made to sample configurational snapshots of single or multiple lattice SFCs. As mentioned above, the algorithm used throughout this work (Fig 1) uses a combination of end attacks and backbites. Nest and replace maneuvers are used to adjust end-to-end scaling of existing chains. These maneuvers can have undesirable effects on (a) the length of each chain, (b) the number of chains, (c) the correlation between snapshots, and (d) the size of the lattice. In what follows we discuss the computational details used to avoid these effects.

a. The length of a chain is prone to alteration with rearrangement by bond flip and end attack. In fact, this is always the case for end attacks. To avoid this effect, we check the length of each chain before outputting computational snapshots for analysis. Any deviation from

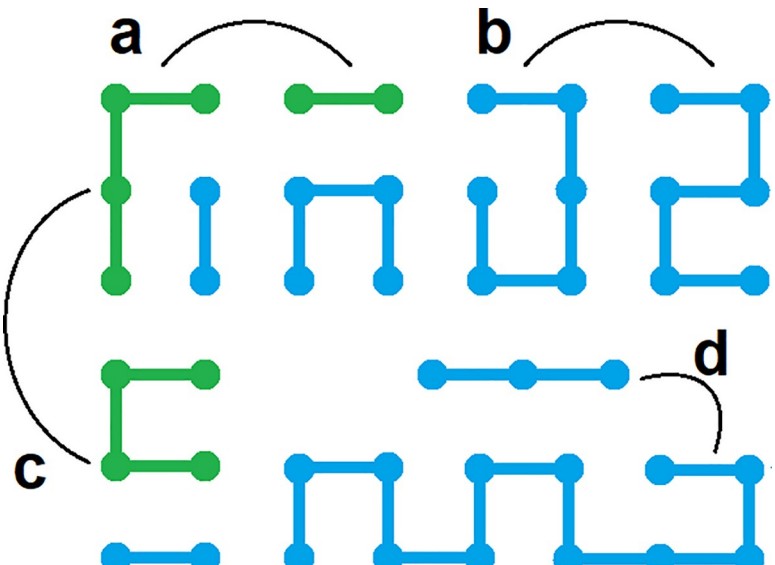

**Fig 8. Schematic of four types of chain rearrangement maneuvers. a,** End attack is an intermolecular chain rearrangement in which a bond is broken and annealed to the end of a different chain. **b,** Backbite is an intramolecular chain rearrangement in which a bond is broken and annealed to the end of the same chain. **c,** Bond flip is an intermolecular chain rearrangement in which two parallel bonds are broken and reformed in perpendicular orientation. **d,** Nest and replace is a rearrangement in which a single monomer is replaced by an entire configurational snapshot.

expected values triggers an additional iteration of the algorithm. Additional checks and iterations are preformed until chain lengths are restored.

b. The number of chains is prone to alteration with rearrangement by end attacks; to some extent this depends on how the maneuver is implemented. In our case this is possible when the attacking endpoint targets another endpoint; consequently, the entire target chain gets appended to the point of attack. To avoid this effect, we prohibit end attacks that target endpoints.

c. End attacks, backbites, and bond flips are only partial rearrangements; parts of the chain(s) are left unchanged. As a result, single rearrangements lead to strongly correlated snapshots. The correlation time has been the subject of previous analysis and is not examined here. To avoid correlated snapshots, our algorithm begins with $10^5$ iterations using a unique random seed. This number is not an arbitrary choice. For 16x16x16 lattices approximately $10^4$ iterations are needed to remove correlations [3]; thus, we safely exceed the minimum number of iterations by an order of magnitude. Additional iterations may or may not be required to correct for deviations in chain length.

d. By design, the nest and replace rearrangement increases the lattice size and length of each chain. These effects are easiest to see with a concrete example. Consider a single lattice SFC on a 4x4x4 lattice (64 points total). Nest and replace each point with 2x2x2 lattice SFC. The result is a single lattice SFC on an 8x8x8 lattice (512 points total). This effect should be anticipated rather than avoided; it is designed to control end-to-end scaling and generate Hilbert-like configurations. Additional details below.

## Details of nest and replace chain rearrangements

The nest and replace rearrangements are executed sequentially–from end to end–for each monomer in a lattice SFC. Each monomer is replaced by a precomputed 2x2x2 lattice SFC drawn from a collection of valid replacements (see Fig 2). Here we discuss how valid replacements are identified and selected.

a. Prerequisite step. Exhaustively compute all 2x2x2 SFCs (as discussed, we do not eliminate paths that are identical by rotation). For the 2x2x2 case there are 144; for the 3x3x3 case there are 4,960,608. Note that the 3x3x3 case is optional in our code (available online) but is not considered in our results.

b. For each monomer in the original lattice SFC (green curve Fig 2), randomly select a valid replacement, i.e., a 2x2x2 lattice SFC where:

   a. The end-wall orientation matches the bond orientation of the monomer (to be replaced) and its neighbor (next to be replaced). See Fig 2.

   b. The start point is adjacent to the endpoint of the previous replacement. See Fig 2.

## Details of Hilbert-like configurations

Hilbert-like configurations of lattice SFCs are generated in two steps. First, an initial configuration is generated at random. Second, each point in the initial path is replaced by an entire configurational snapshot; we refer to these steps as nest and replace maneuvers. In our implementation, valid replacement configurations are drawn at random from the set of all possible 2x2x2 lattice SFCs. Optionally repeat the second step. Each iteration of steps increased the number of points in a lattice SFC by a factor of 8. The new path is Hilbert-like in the sense that its end-to-end scaling has decreased. Three use-case examples provide further clarity.

a. To generate a 4x4x4 Hilbert-like lattice SFC, first generate a 2x2x2 initial configuration (first step above). Then, replace each point in the initial configuration with 2x2x2 lattice SFC (second step above).

b. To generate a 16x16x16 intermediate lattice SFC, first generate an 8x8x8 initial configuration (first step above). Replace each point in the initial configuration with a 2x2x2 lattice SFC (second step above).

c. To generate a 16x16x16 Hilbert-like lattice SFC, first generate a 4x4x4 initial configuration (first step above). Replace each point in the initial configuration with a 2x2x2 lattice SFC (second step above). Repeat the second step once.

## Details of configurational ensembles

Investigation of crossing complexity as an alternative to knot complexity draws from three configurational ensembles: equilibrium, intermediate, and Hilbert-like. All three ensembles consist of snapshots generated on a 16x16x16 lattice. Equilibrium snapshots are generated randomly using the Mansfield algorithm (backbite and end attack maneuvers). Intermediate and Hilbert-like snapshots correspond to the above use-case examples (b) and (c) with slight modifications for S-phase DNA and two-chain configurations (details below).

Investigation of crossing complexity for S-phase DNA begins with three ensembles of configurational snapshots: equilibrium, intermediate (use case example b), and Hilbert-like (use case example c). Each snapshot places one chain (4096 points) on a 16x16x16 lattice. The difference is that each chain is replicated in place akin to S-phase DNA. The "off-lattice" copy is generated in two steps. First, create an exact copy chain superimposed on the original. Second, use a small perturbation–small random translation and rotation–to separate the copy chain from the original while keeping the two juxtaposed in register. The result is a configurational snapshot with two chains, 4096 points each.

Investigation of crossing complexity for multiple paths draws from similar ensembles of configurational snapshots: equilibrium, intermediate (use case example b), and Hilbert-like (use case example c). The difference is that we place two chains on the lattice each with 2048 points. Equilibrium snapshots are generated randomly using the Mansfield algorithm for multiple chains (backbite and end-attack maneuvers). Initial configurations used to generate intermediate and Hilbert-like snapshots begin with two chains but otherwise correspond to use-case examples (b) and (c) detailed above, respectively.

## Test directions

Crossing complexity begins with a set of test directions. Any number of test directions is valid provided they evenly cover the space $S^2$ (spherical surface); we arbitrarily use 1,000 throughout this work. To space points evenly–a difficult problem in $S^2$–we use the Fibonacci method. Details of this method can be found elsewhere [33]. Essentially, points are evenly spaced along a spiral joining two poles of the sphere. The latitude & longitude spherical polar coordinate of point *(n)* in *[1,N]* is given by the formula:

$$P_n = \begin{bmatrix} arcsin\left(n\dfrac{2}{N+1}\right) - 1 \\ n\vartheta \end{bmatrix}$$

Alternative methods for evenly picking points on the spherical surface include lattice points, polar coordinate subdivision, and subdivision of the icosahedron.

### Details of snapshot classification

We quantify the extent to which crossing complexity alone distinguishes configurational snapshots of three computational ensembles: equilibrium, intermediate, and Hilbert-like. Configurational snapshots from all three ensembles (800 snapshots each) are pooled and reclassified one-by-one using a Kolmogorov-Smirnov test. Briefly, a Kolmogorov-Smirnov test is used to compare a one-dimensional probability distribution to a reference probability distribution. In our case, the one-dimensional probability distributions stems from enumerated chain crossings over test directions for each configurational snapshot, i.e. one distribution for each configurational snapshot. There are three reference probability distributions corresponding to the equilibrium, intermediate, and Hilbert-like ensembles, respectively. Reference distributions stem from enumerated chain crossings over test directions pooled for all 800 configurational snapshots in each ensemble. Snapshot distributions are used to perform a test of the null hypothesis; specifically, the sample distribution is drawn from the reference distribution. A null-hypothesis test is performed for all three reference distributions, which produces three p-values. The snapshot classification corresponds to the reference distribution with highest p-value.

### Triple product

The triple product is standard for testing torsion between any set of three unit-vectors: $\vec{a}$, $\vec{b}$, and $\vec{c}$. Briefly, it is defined as the scalar result of $\vec{a} \cdot (\vec{b} \times \vec{c})$. Negative values indicate left-handed torsion while positive values indicate right-handed torsion.

### Details of crossing complexity

Details of crossing complexity can be found elsewhere; nevertheless, we provide a motivating example. We consider the simple case of two straight curves (red and blue line segments in Fig 9A). First, compute a set of test directions that evenly cover the space $S^2$ (100 test directions are shown in Fig 9B). Next, drag the red curve along each test direction counting intersections with the blue curve. Test directions shown in red indicate one crossing; those shown in blue indicate zero crossings (Fig 9B). Statistics of the enumerated chain crossings–such as average and distribution–are used as a measure of entanglement. Crossing complexity is visualized by mapping enumerated chain crossings onto the unit sphere (Fig 9C).

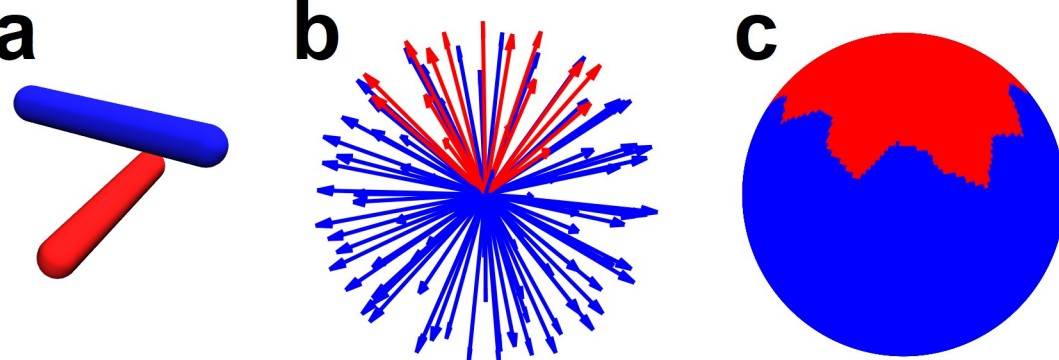

**Fig 9. Crossing complexity visualized for the simple case of two line segments. a,** Two curves shown in red and blue. **b,** Putative translations are applied over a set of test directions that evenly cover the space $S^2$ (spherical surface); chain crossings are enumerated and color coded for each direction. **c,** Enumerated chain crossings mapped onto the unit sphere.

In general, the crossing complexity for two paths (such as the SFCs considered in this work) will have many crossing for each test direction; recall that the paths are separated–to completion–in each direction. Computationally, crossings are enumerated by translating each bond one by one and summing the results for each test direction. See S2 Code.

## Robustness to crossing complexity normalization

A main result of this work hinges on distributions of enumerated chain crossings (crossing complexity) for pairs of equilibrium, intermediate, and Hilbert-like space filing curves (Fig 4A–4C). Differences in those distributions suggest that crossing complexity depends on the folding principles specific to each ensemble (Fig 4D and 4E). However, chain crossings visualized for one snapshot (Fig 3) appear to be anisotropic: we observe relatively few chain crossings in face directions compared to corner directions. We checked robustness of our results by normalizing enumerated crossing numbers by the lattice size measured in the direction of each separation path. The following example provides furthest clarification:

a. Consider a single snapshot consisting of two curves on the same lattice (Fig 10A). Chain crossings are enumerated for test directions that evenly cover the space $S^2$ (Fig 10B). Enumerated chain crossings for ensembles of snapshots produce different distributions depending on how two curves are folded (Fig 10C and 4D).

b. Consider the same chain crossings weighted by the lattice size measured in the direction of each separation path (Fig 10D). Dividing by weight produces a new set of normalized chain crossings (Fig 10E). Normalized chain crossings for the same ensembles–equilibrium,

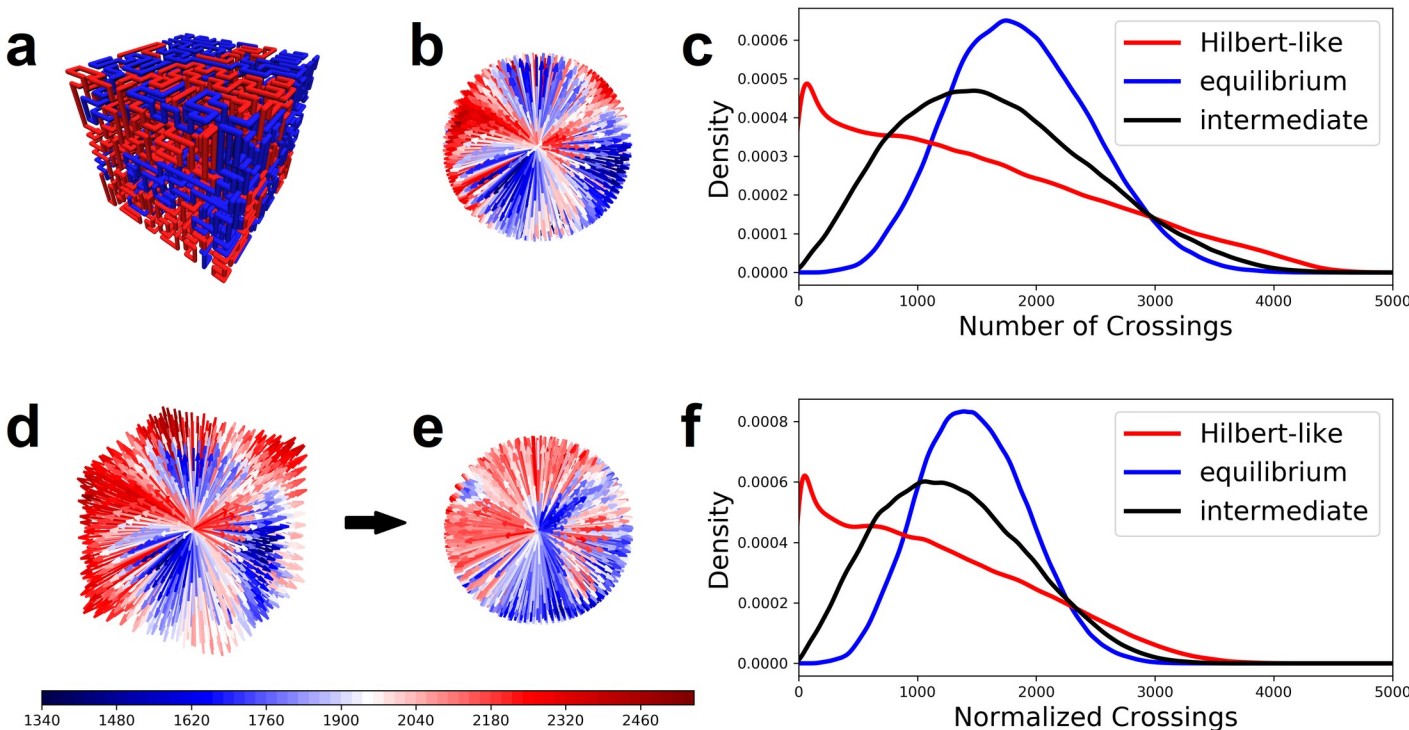

**Fig 10. A key result of this work is robust to crossing complexity normalization. a,** Snapshot of two space filling curves on a lattice. **b,** Chain crossings enumerated for a set of test directions. **c,** The distribution of chain crossings over three different ensembles of configurational snapshots: distributions differ for equilibrium, intermediate and Hilbert-like curves. **d,** Chain crossings enumerated for a set of test directions and weighed by the lattice size measured in the direction of each separation path. **e,** Chain crossings normalized by weight. **f,** Results are robust to crossing complexity normalization.

intermediate, and Hilbert-like–produce different distributions; however, our conclusions are unchanged (Fig 10F). Differences in the distributions suggest that crossing complexity depends on the folding principles specific to each ensemble.

## Supporting information

**S1 Code. Implementation of the space filling curve algorithm used throughout this work.**
(C)

**S2 Code. Implementation of the crossing complexity code used throughout this work.**
(C)

**S1 Fig. Statistics of lattice SFCs suggest they are ergodic.**
(PNG)

## Author Contributions

**Conceptualization:** Nick Kinney, Molly Hickman.

**Data curation:** Nick Kinney, Molly Hickman.

**Formal analysis:** Nick Kinney.

**Funding acquisition:** Nick Kinney, Harold R. Garner.

**Investigation:** Nick Kinney, Molly Hickman.

**Methodology:** Nick Kinney, Ramu Anandakrishnan.

**Project administration:** Nick Kinney, Ramu Anandakrishnan, Harold R. Garner.

**Resources:** Nick Kinney, Ramu Anandakrishnan, Harold R. Garner.

**Software:** Nick Kinney.

**Supervision:** Nick Kinney, Harold R. Garner.

**Validation:** Nick Kinney, Molly Hickman.

**Visualization:** Nick Kinney.

**Writing – original draft:** Nick Kinney, Molly Hickman, Ramu Anandakrishnan, Harold R. Garner.

**Writing – review & editing:** Nick Kinney, Molly Hickman, Ramu Anandakrishnan, Harold R. Garner.

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
