## [Decision Letter · Decision Letter 0]

9 Jul 2020

PONE-D-20-17244

Crossing Complexity of Space Filling Curves Reveals Entanglement of S-Phase DNA

PLOS ONE

Dear Dr. Kinney,

Thank you for submitting your manuscript to PLOS ONE. After careful consideration, we feel that it has merit but does not fully meet PLOS ONE’s publication criteria as it currently stands. Therefore, we invite you to submit a revised version of the manuscript that addresses the points raised during the review process.

Please take into account the referees' suggestions and answer the questions asked by them.

We look forward to receiving your revised manuscript.

Kind regards,

Miguel Angel Sánchez Granero

Academic Editor

PLOS ONE

Journal Requirements:

2.Thank you for stating the following financial disclosure:

 [No, the funders had no role in study design, data collection and analysis, decision to publish, or preparation of the manuscript]

3. Please upload a copy of Supporting Information Figure/Table/etc. S1 Code, S2 Code and Figure S1 which you refer to in your text on page 25.

Reviewers' comments:

Reviewer's Responses to Questions

**Comments to the Author**

1. Is the manuscript technically sound, and do the data support the conclusions?

Reviewer #1: Partly

Reviewer #2: Partly

2. Has the statistical analysis been performed appropriately and rigorously? 

Reviewer #1: N/A

Reviewer #2: I Don't Know

3. Have the authors made all data underlying the findings in their manuscript fully available?

Reviewer #1: Yes

Reviewer #2: Yes

4. Is the manuscript presented in an intelligible fashion and written in standard English?

Reviewer #1: Yes

Reviewer #2: No

5. Review Comments to the Author

Reviewer #1: This paper studies statistical properties of Hamiltonian path pairs in two setups. First, it is the ensemble of two lattice curves whose union passes through every vertex of a finite cubic lattice. Second, a single Hamiltonian path on a cubic lattice is generated and displaced slightly off lattice to represent the second curve. The motivation is the chromatin conformation in eukaryotic nuclei. The first setup is to represent two interphase chromosomes, while the latter one stand for DNA conformation right after duplication. Three different classes of curves are distinguished by their construction method, for each class an ensemble of conformation is simulated and the crossing number distribution is computed.

While the work has some interesting aspects, some parts lack necessary detail, some other parts may benefit from more insight from the results already reported in the literature.

Here are some major issues in detail:

(1) The terminology used in the introduction and abstract for the different curves is a bit confusing. The authors should try to clarify more sharply the distinction of ``fractal", ``space-filling", ``Hamiltonian" and ``equilibrium" curves and their relation to the presence of knots. Firstly, ``fractal" does not necessarily mean the curves don't have knots, but simply that they are self-similar. Ordinary random walk is also fractal, but exhibits knots when computed on properly closed curve [Meyer et al, ACS Macro Lett. 2018, 7, 6, 757--761]. Second, ``space-filling" typically denotes curves with fractal dimension $d$, i.e. that of the embedding space. In other words, the exponent $\\alpha$ (commonly named $\\nu$) having the value of $1/d$ where $d$ is the space dimension. Based on the constructions in [Smrek, Grosberg, Physica A 2013, 392 (24), 6375-88], these curves could also be knotted. Third, connection to ``Hamiltonian" paths i.e. ones visiting every vertex of a graph just once should be made. Fourth, ``equilibrium" curves should be probably discussed from the polymer melt perspective [Meyer et al above]. Fifth, a brief explanation of what is meant by knots on open curves would help the reader. Sixth, another useful paper in the context of chromatin and knottedness is [Goundaroulis et al, Biophysical Journal, 118, 9, 5 2020, 2268--2279]. A deeper exposure of the differences in the chromatin context could be found in [Halverson et al 2014 Rep. Prog. Phys. 77 022601].

(2) The latter reference, besides the exponent $\\alpha$ ($\\nu$), discusses also exponent $\\beta$ that characterizes fractal dimension of the curve's boundary for the SFCs or the scaling of the number of contacts between two curves. This seems strongly related to the crossing number, which presumably characterizes how intermixed the two curves in question are. It would be useful to discuss the results in this context.

(3) Fig3: It simply looks like there are fewer chain crossings in the face directions in comparison to the corner directions. This is no surprise, as the latter has more sites by a factor of $\\sqrt{3}$, which agrees with the scale in (e): $1340 \\sqrt{3}=2320$. In this sense it would be useful to normalize the net crossing numbers in a given direction by the length of the separation path in a given direction, and compute the respective distribution only afterwards.

(4) pg10 ln 153: ``fractal curves are easy to pull apart". Not all of them, it likely depends on $\\beta$. It looks like here by fractal curves the author mean the traditional ones with smooth surface i.e. $\\beta=(d-1)/d$

(5) pg10 ln 156-161: The classification of the curves to the three classes is artificial, because equilibrium class can contain also fractal-like one.

(6) pg10 ln164 -165: ``Physical interpretation of the crossing complexity is that chains are easiest to pull apart in directions with few accumulated crossings (figure 3)". Is this a result of a simulation, experiment or just an intuitive idea? This should be made clear.

(7) On the s-phase model: There should be a geometric argument why the doubled curves show the same crossing number distributions.

(8) On the ergodicity discussion. It is stated correctly that the indications suggest, but do not prove unbiased sampling. Would it be possible to test this on smaller cases where the total enumeration is possible? "Small" Hamiltonian paths are enumerated in [Schram and Schiessel, 2013 J. Phys. A: Math. Theor. 46 485001] while some smooth fractal curves in [Smrek and Grosberg 2015 J. Phys. A: Math. Theor. 48 195001]. Besides, is it trivially obvious that the endpoints positions of the fractal curves must be uniformly distributed within the cube (pg14 ln229)? From the latter paper this does not seem to be so.

(9) Correlation times of the curves' generation algorithms are not discussed, only some number is mentioned on ln363. How do we know the samples are uncorrelated?

In summary, this is potentially an interesting paper on the statistical properties of sub-classes of Hamiltonian paths, but significant clarification and much more serious connection to the literature is absolutely necessary.

Reviewer #2: Space-filling curves (SFC's) constitute a special class of mathematical objects describing the behavior of compact polymers: two major sub-classes include equilibrium SFC's and crumpled SFC's. In particular the latest are self-similar, displaying the same fractal behavior at all spatial scales, and minimally entangled and hence have been employed to describe the physical properties of naturally-occurring compact polymers like globular proteins and chromosomes.

The work by Kinney et al. aims at characterizing the topological properties of SFC's by applying the concepts of knot complexity vs. crossing complexity: the first variable quantifies the presence of knots inside the polymer chain while the second measures the amount of chain crossings between any two chain strands which are directed against each other along some randomly-chosen spatial orientation. The two quantities are applied on compact conformations of equilibrium and fractal polymers: Kinney et al. have produced these conformations generalizing a lattice algorithm whose first implementation was due to Mansfield (Ref. [2] of this work).

The generalization of Mansfield's algorithm is one of the motivations behind the present work, and I think it is an interesting one. Unfortunately, concerning the rest of the paper, I am much less positive and I am now trying to explain why:

(1) Obviously, the main message of this work is that since knottedness is not sufficient to discriminate between equilibrium and fractal curves, chain crossing is proposed as a better indicator (Fig. 4): intuitively, deciding that two compact curves may or may not be knotted (based on some knot invariant) could indeed be complicate so I find reasonable what the authors say. Unfortunately, I would have preferred to see their claim motivated by quantitative analysis, while they only point to an obscure (at least for me!) reference (Ref. [12], author: Golyk VA) which is not available. I find this unfair, references must be available either published or at least in preprint form.

(2) But even before my point (1) I have an even more serious concern: I suppose that the polymers chains simulated by Kinney et al. are linear, open polymers. Rigorously, knots exist only for closed curves (rings): yet, knots can still be generalized to open curves provided some numerical "tricks" or definition are adopted (see for instance: Micheletti, Marenduzzo, Orlandini, Phys. Rep. (2011)). Do the authors look into this literature? I think they have to: these "tricks" are relatively standard now, and the authors should analyze their curves based on these methods. Only after that, the comparison to decide which indicator between knottedness and crossing performs better would reveal its true potential.

3) Honestly, I am missing the message on the S-phase DNA model (Fig. 5): true, there crossing complexity can not distinguish between equilibrium and fractal models, but I think this is just because the model consists of a new chain (say, the red one) which is built to run in parallel with the blue one. Chain crossing is based on local moves so I think any two chains running in parallel (regardless of the details of their global folding in space) should always produce the same result.

To summarize, this paper raises several serious concerns which require some substantial reply by the authors should they wish to resubmit their work to Plos One.

6. PLOS authors have the option to publish the peer review history of their article (what does this mean?). If published, this will include your full peer review and any attached files.

Reviewer #1: **Yes: **Alexander Y Grosberg and Jan Smrek

Reviewer #2: No

---

## [Author Response · Author response to Decision Letter 0]

25 Jul 2020

[all responses to review comments are in the attached document "Response to Reviewers.pdf"]

---

## [Decision Letter · Decision Letter 1]

7 Aug 2020

PONE-D-20-17244R1

Crossing Complexity of Space Filling Curves Reveals Entanglement of S-Phase DNA

PLOS ONE

Dear Dr. Kinney,

Thank you for submitting your manuscript to PLOS ONE. After careful consideration, we feel that it has merit but does not fully meet PLOS ONE’s publication criteria as it currently stands. Therefore, we invite you to submit a revised version of the manuscript that addresses the points raised during the review.

Please, address the remaining questions posed by one of the reviewers. 

We look forward to receiving your revised manuscript.

Kind regards,

Miguel Angel Sánchez Granero

Academic Editor

PLOS ONE

Reviewers' comments:

Reviewer's Responses to Questions

**Comments to the Author**

1. If the authors have adequately addressed your comments raised in a previous round of review and you feel that this manuscript is now acceptable for publication, you may indicate that here to bypass the “Comments to the Author” section, enter your conflict of interest statement in the “Confidential to Editor” section, and submit your "Accept" recommendation.

Reviewer #1: (No Response)

Reviewer #2: All comments have been addressed

2. Is the manuscript technically sound, and do the data support the conclusions?

Reviewer #1: Yes

Reviewer #2: Yes

3. Has the statistical analysis been performed appropriately and rigorously? 

Reviewer #1: I Don't Know

Reviewer #2: Yes

4. Have the authors made all data underlying the findings in their manuscript fully available?

Reviewer #1: Yes

Reviewer #2: Yes

5. Is the manuscript presented in an intelligible fashion and written in standard English?

Reviewer #1: Yes

Reviewer #2: Yes

6. Review Comments to the Author

Reviewer #1: Authors in many places made a good job responding to the critical remarks from the report. but in some places their presentation still needs polishing.

1) Authors start defining space-filling curves as "configurations with fractal dimension equal to the embedding space". Apart from clumsy English (dimension equal ... space), this definition is wrong. For instance, one can easily imagine a space filling curve in 2D with fractal dimension 1 -- a tight spiral starting from zero and filling a circle.

2) Some statements are also unclear, such as, e.g., "the 3D distance between any two monomers (R) cannot exceed their 2D separation along the polymer backbone" For is 2D separation? Authors continue "In polymer physics this constraint is expressed as an inequality (R <= s)..." and accompany this with reference to de Gennes. I don't remember such statement in de Gennes book; if it is there, authors should be more specific in their referencing.

3) Around line 250, authors claim that ergodic algorithm should produce torsion-free curves. Why? It is not obvious to me.

4) There is a chapter called "STATISTICS OF SPACE FILLING CURVES SUGGEST THEY ARE EGODIC" Curves are ergodic? What does it mean? I never heard of ergodicity viewed as a property of a curve.

Reviewer #2: The authors have satisfactorily addressed my concerns, the present version of the work is much more clear than the original submission. Therefore, I recommend the publication without further hesitation.

7. PLOS authors have the option to publish the peer review history of their article (what does this mean?). If published, this will include your full peer review and any attached files.

Reviewer #1: No

Reviewer #2: No

---

## [Author Response · Author response to Decision Letter 1]

11 Aug 2020

Reviewer #1

Reviewer #1: Authors in many places made a good job responding to the critical remarks from the report. but in some places their presentation still needs polishing.

We thank the reviewer for the helpful comments.

Authors start defining space-filling curves as "configurations with fractal dimension equal to the embedding space". Apart from clumsy English (dimension equal ... space), this definition is wrong. For instance, one can easily imagine a space filling curve in 2D with fractal dimension 1 -- a tight spiral starting from zero and filling a circle.

This passage was an oversight. We were probably referring to classical curves such as Peano, Hilbert, and Moore space filling curves. Obviously, our work considers equilibrium and intermediate configurations that render this passage incorrect. In fact, it may be possible to relate fractal dimension to crossing complexity. We now elude to this possibility in the discussion where we also suggest a relationship between crossing complexity and surface smoothness (β). It would be interesting to see a thorough comparison of these measurements; however, that will have to wait for another study.

Some statements are also unclear, such as, e.g., "the 3D distance between any two monomers (R) cannot exceed their 2D separation along the polymer backbone" For is 2D separation? 

This was another oversight. The sentence should read as follows. For example, the 3D distance between any two monomers (R) cannot exceed their 1D separation along the polymer backbone (s).

Authors continue "In polymer physics this constraint is expressed as an inequality (R <= s)..." and accompany this with reference to de Gennes. I don't remember such statement in de Gennes book; if it is there, authors should be more specific in their referencing.

The inequality (R <= s) is implied but never explicitly stated. In retrospect, it is implicit in any discussion of polymers which makes a specific citation inappropriate. We have tweaked the wording of the sentence and removed the citation. The de Gennes book is still cited at the end of the paragraph.

Around line 250, authors claim that ergodic algorithm should produce torsion-free curves. Why? It is not obvious to me.

Actually, the statement we make – that chain configurations are torsion free – is inaccurate. Just look at figure 7 panel b and c. Most configurations have a degree of torsion. So what do we expect in an unbiased sample of configurations? Consider a hypothetical lattice configuration with a degree of right-handed torsion. There must be a “mirror image” configuration with the same degree of left-handed torsion. In fact, for every configuration with right-handed torsion there should be a configuration with equal left-handed torsion. Thus, we expect the distribution of torsion in an unbiased ensemble to be symmetric (not necessarily bell-shaped) around zero. We have appended this argument to the passage around line 250. 

There is a chapter called "STATISTICS OF SPACE FILLING CURVES SUGGEST THEY ARE EGODIC" Curves are ergodic? What does it mean? I never heard of ergodicity viewed as a property of a curve.

This was a poor choice of words. We change the section heading to “EGODIC SAMPLING OF LATTICE SPACE FILLING CURVES”

---

## [Decision Letter · Decision Letter 2]

14 Aug 2020

Crossing Complexity of Space Filling Curves Reveals Entanglement of S-Phase DNA

PONE-D-20-17244R2

Dear Dr. Kinney,

We’re pleased to inform you that your manuscript has been judged scientifically suitable for publication and will be formally accepted for publication once it meets all outstanding technical requirements.

Kind regards,

Miguel Angel Sánchez Granero

Academic Editor

PLOS ONE

Additional Editor Comments (optional):

Reviewers' comments:

Reviewer's Responses to Questions

**Comments to the Author**

1. If the authors have adequately addressed your comments raised in a previous round of review and you feel that this manuscript is now acceptable for publication, you may indicate that here to bypass the “Comments to the Author” section, enter your conflict of interest statement in the “Confidential to Editor” section, and submit your "Accept" recommendation.

Reviewer #1: All comments have been addressed

2. Is the manuscript technically sound, and do the data support the conclusions?

Reviewer #1: Yes

3. Has the statistical analysis been performed appropriately and rigorously? 

Reviewer #1: N/A

4. Have the authors made all data underlying the findings in their manuscript fully available?

Reviewer #1: Yes

5. Is the manuscript presented in an intelligible fashion and written in standard English?

Reviewer #1: Yes

6. Review Comments to the Author

Reviewer #1: (No Response)

7. PLOS authors have the option to publish the peer review history of their article (what does this mean?). If published, this will include your full peer review and any attached files.

Reviewer #1: No

---

## [Editor Report · Acceptance letter]

18 Aug 2020

PONE-D-20-17244R2 

Crossing Complexity of Space Filling Curves Reveals Entanglement of S-Phase DNA 

Dear Dr. Kinney:

I'm pleased to inform you that your manuscript has been deemed suitable for publication in PLOS ONE. Congratulations! Your manuscript is now with our production department. 

Kind regards, 

on behalf of

Dr. Miguel Angel Sánchez Granero 

Academic Editor

PLOS ONE